

# Geographic variation and core microbiota composition of *Anastrepha ludens* (Diptera: Tephritidae) infesting a single host across latitudinal and altitudinal gradients

Martín Aluja[1,*], Daniel Cerqueda-García[1,*], Alma Altúzar-Molina[1], Larissa Guillén[1], Emilio Acosta-Velasco[1], Juan Conde-Alarcón[1] and Andrés Moya[2]

[1] Red de Manejo Biorracional de Plagas y Vectores, Clúster Científico y Tecnológico BioMimic, Instituto de Ecología, A.C.–INECOL, El Haya, Xalapa, Veracruz, Mexico
[2] Instituto de Biología Integrativa de Sistemas (I2SysBio), Universidad de Valencia y Consejo Superior de Investigaciones Científicas (CSIC), Valencia, Spain
* These authors contributed equally to this work.

Corresponding authors
Martín Aluja,
martin.aluja@inecol.mx
Daniel Cerqueda-García,
daniel.cerqueda@inecol.mx

## ABSTRACT

*Anastrepha ludens* is a pestiferous tephritid fly species exhibiting extreme polyphagy. It develops optimally in hosts rich in sugar but low nitrogen content. We studied the geographical influence on the composition of *A. ludens*'s larval and newly emerged adult gut microbiota in altitudinal (0–2,000 masl) and latitudinal (ca. 800 km from 17° to 22°N latitude) transects along the coastline of the state of Veracruz, Mexico. In the 16 collection sites, we only collected *Citrus x aurantium* fruit (238 samples of *A. ludens* larvae and adults, plus 73 samples of pulp) to control for host effect, hypothesizing that there exists a conserved core microbiota that would be dominated by nitrogen-fixing bacteria. We found that latitude triggered more significant changes in the gut microbiota than altitude. Northern and southernmost samples differed the most in microbiota composition, with a trade-off between Acetobacteraceae and Rhizobiaceae driving these differences. As hypothesized, the core microbiota in each sampling site, contained the functional group of nitrogen-fixing bacteria. We conclude that *A. ludens* larvae can acquire multiple diazotrophic symbionts along its wide distribution range where it infests fruit with a high C:N ratio in the pulp.

## INTRODUCTION

Insects constitute a significant proportion of the Earth's biomass, surpassing any other animal group (*Bar-On, Phillips & Milo, 2018*). Remarkable examples are the termites, whose contribution to the overall biomass surpasses that of all vertebrate classes combined (*Bar-On, Phillips & Milo, 2018*; *Turner, 2019*). Many insects like termites have phytophagous habits, feeding on diets with low nitrogen content, such as fruit pulp, leaves,

and wood. These insects with a high carbon/nitrogen (C:N) ratio diets overcome nitrogen scarcity through their microbiota (*Bar-Shmuel, Behar & Segoli, 2020*; *Ren et al., 2022b*). This supplementation of nitrogen by the microbiota has been substantiated through both direct and indirect evidence across a broad taxonomic range of insects, such as Coleoptera, Hymenoptera, Hemiptera, and Diptera (*Ceja-Navarro et al., 2014*; *De León et al., 2017*; *Bar-Shmuel, Behar & Segoli, 2020*). In the case of termites, their gut microbiota contains *Klebsiella* and *Enterobacter* which aid in nitrogen supplementation by biological nitrogen fixation (BNF henceforth) (*Potrikus & Breznak, 1977*; *Ulyshen, 2015*).

Within the microbiota of Diptera, BNF has been confirmed in Tephritids, specifically in the Queensland fruit fly *Bactrocera tryoni* (Froggatt) (*Murphy, Teakle & MacRae, 1994*), the Mediterranean fruit fly *Ceratitis capitata* (Wiedemann) (*Behar, Yuval & Jurkevitch, 2005*; *Behar, Jurkevitch & Yuval, 2008*), and the Oriental fruit fly *Bactrocera dorsalis* (Hendel) (*Ren et al., 2022a*). *Behar, Yuval & Jurkevitch (2005, 2008)* highlighted the role that diazotrophic enterobacteria vertically transmitted from females to their offspring have in facilitating the acquisition of nitrogen from rotting fruit pulp.

The Mexican fruit fly, *Anastrepha ludens* (Loew) (Diptera: Tephritidae), our study model here, is a polyphagous species that ancestrally infests white sapote (*Casimiroa edulis* La Llave & Lex), a fruit with high sugar but negligible nitrogen content, representing a nutritional source with a high carbon/nitrogen (C:N) ratio (*Morton, 2013*; *Birke & Aluja, 2018*). The domestication of several wild fruits has selected for and fixed this trait, as fruits become more palatable to humans (*i.e.*, higher sugar content), thereby increasing their commercial value. In contrast, wild fruits typically contain high concentrations of secondary metabolites that confer an acidic or bitter taste, such as flavonoids, tannins, and other polyphenols, including catechins, prunins and rutins (*Aluja & Mangan, 2008*; *Meyer, DuVal & Jensen, 2012*; *Aluja et al., 2014*; *Zhang & Hao, 2020*; *Dar et al., 2021*; *Cao et al., 2022*). However, increasing sugar content and lowering the levels of secondary metabolites, such as polyphenols, also renders the domesticated fruit more suitable for the development of the larvae of *A. ludens*. This is because their high sugar content resembles that of the ancestral host *C. edulis*, making commercially grown fruit more vulnerable to higher levels of infestation by this pestiferous frugivore. Thus, domesticated fruits could represent a shuttle in the expansion of *A. ludens* to new latitudes and host plants (*Aluja et al., 2014*).

The polyphagous habit of *A. ludens* lacks a phylogenetic-specific host-fly relationship (*Birke & Aluja, 2018*; *Ochoa-Sánchez et al., 2022*). However, in nature this fly cannot infest fruit with high levels of tannins (*Birke, Acosta & Aluja, 2015*; *Birke & Aluja, 2018*). In a recent study, we observed that the larvae of this species exhibit a poor or deleterious development when forced to infest guavas (*Ochoa-Sánchez et al., 2022*), a non-host fruit with a significantly high polyphenol content compared with the ancestral host white sapote, *C. edulis* (*Shabbir et al., 2020*; *Abo Taleb & Abdul Latif, 2023*). *Ochoa-Sánchez et al. (2022)* compared the gut microbiota of two species that thrive in guava in nature (*i.e.*, *Anastrepha striata* and *Anastrepha fraterculus*) with the one of *A. ludens*, and discovered that the guts of *A. striata* and *A. fraterculus* larvae contained the acetic acid bacterium (Acetobacteraceae) *Komagataeibacter* in significant abundance, contrary to *A. ludens*, in which case the gut of larvae contained this microorganism in negligible abundances or

lacked it (*Aluja et al., 2021*; *Ochoa-Sánchez et al., 2022*; *Salgueiro et al., 2022*). But *A. ludens* harbors other Acetobacteraceae genera in its gut microbiota, such as *Acetobacter* and *Gluconobacter* (*Aluja et al., 2021*; *Ochoa-Sánchez et al., 2022*). Both genera seem to be crucial for the correct development of *A. ludens* larvae, but they are depleted when *A. ludens* is forced to develop in guavas or the marginal host *Capsicum pubescens* R&P (*Aluja et al., 2021*). Importantly, *Acetobacter* and *Gluconobacter* are known to metabolize sugars and alcohol with a high rate of BNF (*Reis & Teixeira, 2015*; *Pedraza, 2016*).

The bitter orange, *Citrus* x *aurantium* L., a preferred host of *A. ludens* throughout its distribution range (*Aluja & Mangan, 2008*), is a fruit with a high C:N nutritional ratio and is distributed along a broad range of altitudinal (0–2,000 masl) and latitudinal (17° to 22°N) sites within the Mexican State of Veracruz. This broad geographic distribution renders *C. x aurantium* an ideal model host to characterize the changes in the microbiota of *A. ludens* related to geographic variables and to ascertain if this species contains a core microbiota, something that Aluja and collaborators (*Aluja et al., 2021*) were not able to determine when comparing the gut microbiota in larva originating in six hosts belonging to four plant families. Recent studies suggest that geographical factors such as altitude and latitude can significantly influence microbiota composition due to environmental gradients like temperature and humidity (*Schemske et al., 2009*; *Sepulveda & Moeller, 2020*). For instance, microbial diversity in insect guts could be shaped by local environmental conditions, with altitude and latitude driving differences in both host plants and associated insect species (*Yu et al., 2021*; *Zhang et al., 2023*; *Yang et al., 2024*; *Zhang et al., 2024*). Based on the general postulates of *Bar-Shmuel, Behar & Segoli (2020)*, and considering the fact that the pulp of *C. x aurantium* is rich in sugars but poor in protein (*i. e.*, ratio of 18.4 (16.6%/0.9%)) (*Morton, 2013*; *Rivera, Bocanegra-García & Monge, 2010*), we sampled and analyzed the gut microbiota of *A. ludens* infesting *C. x aurantium* in locations covering the entire altitudinal and latitudinal range of *C. x aurantium* within the Mexican state of Veracruz, using an amplicon sequencing approach. We hypothesized that altitude would exert a greater influence than latitude on the microbiota composition of *A. ludens* due to the significant temperature variation from the coastline to mountainous biomes. Additionally, we hypothesized that the core microbiota would be likely dominated by nitrogen fixing bacteria (NFB henceforth) to compensate for the lack of nitrogen in the feeding substrate of the larvae.

## MATERIALS AND METHODS

### Biological material

To obtain third instar larvae and newly emerged *A. ludens* adults, naturally infested bitter oranges (*C.* x *aurantium*) were collected along latitudinal and altitudinal gradients in the State of Veracruz, Mexico. Veracruz has an elongated shape with a length of approximately 790 km from N to S measured along the coastline. But since it is also situated along the Sierra Madre Oriental (except for the most southern part), one can find many locations all the way from sea level to above 2,000 m. Taking advantage of this unique opportunity, in the case of the latitudinal gradient, we sampled in six sites located between 17° and 22°N (Table 1 and Fig. 1). In the altitudinal gradient, ten sites were sampled from 0 to 2,000

**Table 1 Sampling sites along an latitudinal and altitudinal gradient in the state of Veracruz, Mexico to collect naturally infested bitter orange fruit (*C. x aurantium*) as a source of larvae to study their gut microbiota.**

| Sampling range | | Locality | Municipality | Sampling date | Latitude (N) | Longitude (W) | Altitude (masl) |
|---|---|---|---|---|---|---|---|
| **Latitudinal gradient** | 17–18°N | El Chichón | Las Choapas | 17-02-2017 | 17°45′17.87 | 94°6′42.55 | 80 |
| | 18–19°N | Santiago Tuxtla | Santiago Tuxtla | 17-02-2017 | 18°27′39.97 | 95°17′45.94 | 252 |
| | 19–20°N | Martínez de la Torre | Martínez de la Torre | 20-03-2017 | 19°59′36.43 | 96°52′55.00 | 67 |
| | 20–21°N | Zapotal | Tihuatlán | 17-03-2017 | 20°51′02.5 | 97°28′53.9 | 15 |
| | 21–22°N | San Juan Casiano | Chinampa de Gorostiza | 17-03-2017 | 21°23′09.2 | 97°40′33.1 | 110 |
| | 22–23°N | Km 80 | Tampico el Alto | 18-03-2017 | 22°00′08.03 | 97°46′33.3 | 48 |
| **Altitudinal gradient** | 0–200 masl | Armadillo-Cardel | Puente Nacional | 27-02-2017 | 19°21′13.20 | 96°29′55.75 | 147 |
| | 200–400 masl | Apazapan | Apazapan | 24-02-2017 | 19°19′21.17 | 96°43′7.64 | 343 |
| | 400–600 masl | Las Cumbres | Emiliano Zapata | 23-03-2017 | 19°22′43.10 | 96°40′27.53 | 435 |
| | 600-800 masl | Sur Costa Rica | Tuzamapan | 13-03-2018 | 19°23′29.01 | 96°51′40.96 | 687 |
| | 800–1,000 masl | Bella Esperanza | Coatepec | 21-03-2017 | 19°25′39.36 | 96°51′33.05 | 980 |
| | 1,000–1,200 masl | El Mohón* | Hueytamalco | 21-03-2017 | 19°54′35.69 | 97°16′52.10 | 1,021 |
| | 1,200–1,400 masl | Veracruz-Puebla | Tlapacoyan | 21-02-2017 | 19°53′8.38 | 97°17′56.13 | 1,295 |
| | 1,400–1,600 masl | Tlalnelhuayocan | Tlalnelhuayocan | 21-02-2017 | 19°33′43.65 | 96°58′16.14 | 1,569 |
| | 1,600–1,800 masl | Tlalnelhuayocan | Tlalnelhuayocan | 23-02-2017 | 19°33′57.89 | 96°58′23.35 | 1,642 |
| | 1,800–2,000 masl | Tlalnelhuayocan | Tlalnelhuayocan | 03-11-2019 | 19°34′21.49 | 96°59′52.83 | 1,875 |

**Notes:**
* Located in the border between the states of Veracruz and Puebla, Mexico.

meters above sea level in the central part of the same state (*i.e.*, same latitude; Table 1). Five fruits were collected randomly in each collection site for larval sampling, and additional fruit was sampled to obtain adults (details follow). Only fruit with signs of infestation (soft spots when skin was pressed) but without wounds from birds or insects were collected. Once collected, infested fruit were transported to the headquarters of the Instituto de Ecología, A.C.—INECOL for processing. As we worked with a notorious agricultural pest, neither ethical nor collection permits were necessary.

## Sample processing

Two larvae and around 200 mg of "clean" pulp surrounding the larvae were sampled per infested fruit under aseptic conditions (ten larvae from five fruits per collection site). The clean pulp had no vestiges of larvae or larval activity. Pulp was frozen in liquid nitrogen, and larvae were washed superficially and individually according to Aluja et al. (*Aluja et al., 2021*). First, larvae were washed with 500 μL of washing solution (1% SDS, 10 mM Tris pH 8.0, 10 mM NaCl) for 1 minute. Then, 1 minute with 500 μL of 1% commercial bleach; 1 minute with 500 μL of 70% ethanol, and finally, two washes (1 minute each) with sterile distilled water. Then, mechanical pressure with a sterile pestle was applied to the washed larvae to separate the cuticle from the internal content. The internal contents of the larvae (mainly the gut with some fatty tissue) and "clean" pulp were stored at −80 °C until DNA extraction procedures began. The internal contents of five larvae and "clean" pulp

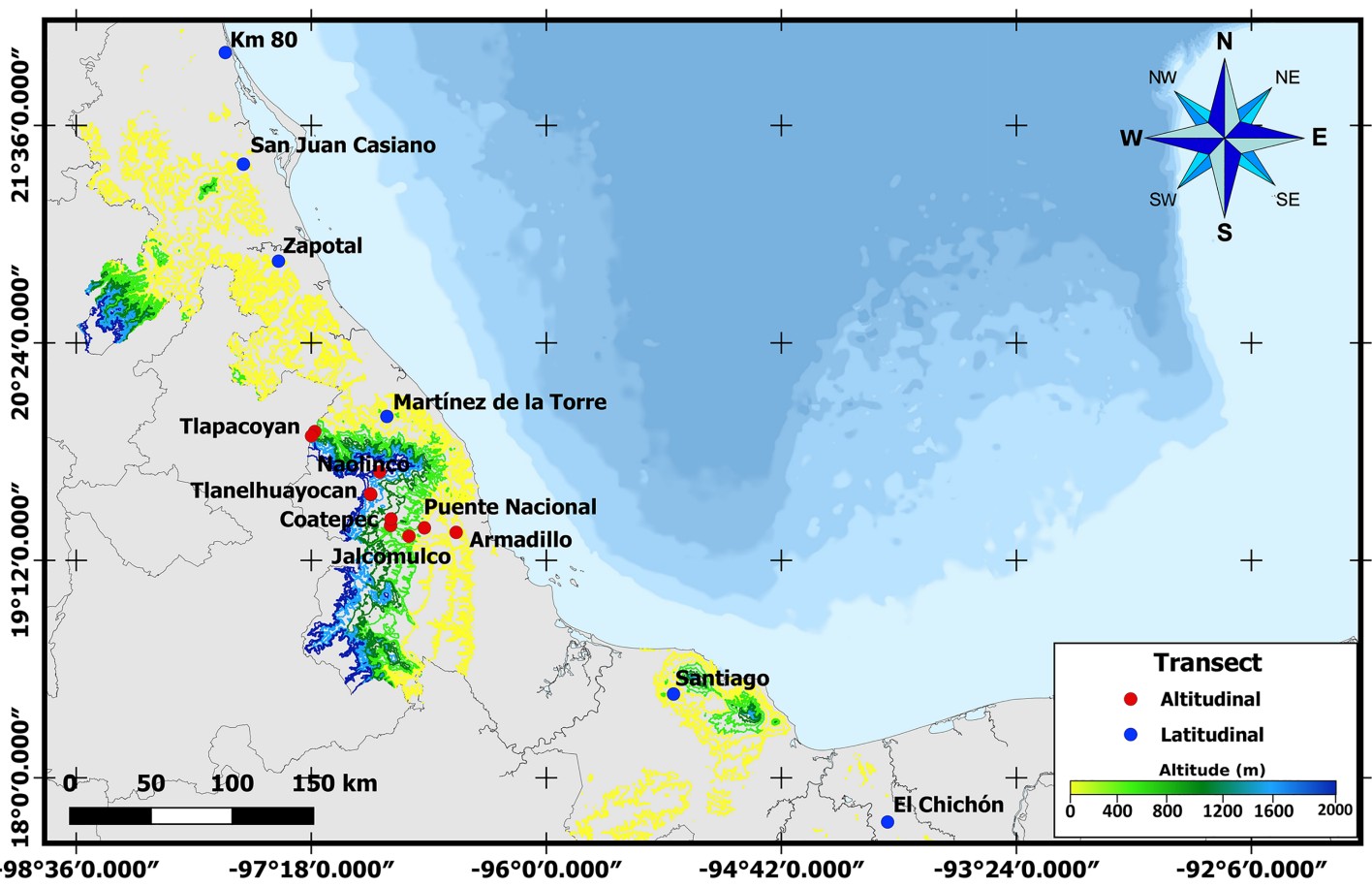

**Figure 1 Sampling sites across Veracruz.** Map displaying locations where samples were collected along altitudinal (marked in red) and latitudinal (marked in blue) transects throughout the Mexican state of Veracruz.

stemming from five different fruits per collection site were used for DNA extraction. We note that we did not thoroughly dissect the gut of the larvae (*i.e.*, some fatty tissue remained attached to the gut), as based on *De Cock et al. (2019)* dissection technique had no major effect on microbiota composition in the case of *C. capitata* using high-throughput sequencing techniques. Also, since here we only concentrated on the 16S gene, as opposed to following a metagenomic approach.

For adult sampling, infested fruit (5–10) from each site collected in addition to the fruit used to extract larvae (described above), were placed in baskets (32.5 cm × 26.5 cm × 8.5 cm) on plastic trays (36 cm × 30 cm × 14 cm) containing a thin layer of sterile vermiculite at the bottom for pupation and covered with pantyhose cloth to avoid the contact with other flies. Fly pupae were recovered daily, transferred to plastic containers (250 mL) with sterile vermiculite, and moistened with sterile distilled water each three days until adult emergence. Newly emerged adults were sacrificed and washed immediately as was the case with larvae. Then, the gut (from cardia to the anus) was dissected and stored with RNA later at −80 °C until DNA processing. Ten guts of females and males (two per sample in each of the five fruits (replicates)) were used for DNA extraction.

## DNA isolation, 16S rRNA gene amplification and sequencing

Individual samples consisted of 20 mg of fruit pulp, the gut and some attached tissue (*e.g.*, fatty tissue) of one larva, and, separately, two guts of males and females (in this case we used pools of two guts per sex as DNA extraction of a single gut was weak). As noted before, there were five replicates per collection site. DNA was isolated using the QIamp® DNA Mini kit (Qiagen©, Hilden, Germany) following the instructions of the manufacturer. DNA was quantified with a spectrophotometer Biospec-nano (Shimadzu®, Kyoto, Japan). Five replicates of each tissue were analyzed per collection site. The 16S rRNA gene was amplified using the primers with overhang adapter sequence Forward 5′TCGTCGGCAGCGTCAGATGTGTATAAGAGACAGCCTACGGGNGGCWGCAG and Reverse 5′GTCTCGTGGGCTCGGAGATGTGTATAAGAGACAGG ACTACHVGGGTATCTAATCC, covering the V3-V4 hyper variable regions, according to *Klindworth et al. (2013)* and the 16S metagenomic sequencing library preparation guide (https://emea.illumina.com/content/dam/illumina-support/documents/documentation/chemistry_documentation/16s/16s-metagenomic-library-prep-guide-15044223-b.pdf). For amplification, a nested polymerase chain reaction (PCR) was performed according to *Ochoa-Sánchez et al. (2022)*. The PCR1 (25 μL) consisted of 100 ng of DNA, Qiagen buffer 1×, $MgCl_2$ 0.1 μM, dNTPs 0.2 μM, forward & reverse primers 0.2 μM, and Taq polymerase 0.05 U, at 1 cycle: 94 °C/ 2 min, 15 cycles: 94 °C/15 s, 55 °C/ 30 s, 72 °C/ 1 min, and 1 cycle: 72 °C/ 5 min. For PCR2 (100 μL) we used two μL of PCR1 product and same reaction conditions used for PCR1, except that in PCR2 we used 25 cycles of PCR amplification. Amplicons were purified using the Promega Wizard® SV gel and PCR clean-up system (Promega Corp., Madison, WI, USA). Purified amplicons were sequenced by Macrogen Inc. (Seoul, Rep. of Korea) using the Illumina Miseq (300 bp) platform.

## Bioinformatic and statistical analyses

We processed the paired-end Illumina raw reads (2 × 250) with the QIIME2 pipeline (*Bolyen et al., 2019*). Amplicon sequence variants (ASVs) were resolved with the DADA2 plugin (*Callahan et al., 2016*) considering the following parameters: –p-trim-left-f 20, –p-trim-left-r 20, –p-trunc-len-f 260, –p-trunc-len-r 240, –p-chimera-method pooled. The representative sequences of ASVs were taxonomically classified with the classify-consensus-vsearch plugin (*Rognes et al., 2016*) using the SILVA database (v138) as a reference (*Quast et al., 2013*). We built a phylogenetic tree with the representative sequences of ASVs with the align-to-tree-mafft-fasttree (*Katoh, 2002*; *Price, Dehal & Arkin, 2010*). All output data were exported to the R environment (*R Development Core Team, 2021*) using the phyloseq package (*McMurdie & Holmes, 2013*).

In R, we partitioned the samples into two types: those derived from the fly's microbiota and those from the clean pulp microbiota. ASVs corresponding to chloroplasts and mitochondria were discarded, and the sample size was normalized using the SRS (Scalling Ranked Subsampling) package (*Beule & Karlovsky, 2020*) to 5,000 and 500 counts for the fly and pulp microbiota, respectively. Beta diversity was calculated using unweighted UniFrac distance. A Gap statistics analysis was performed on the distance matrix to determine the optimal sample clusters, utilizing the cluster library (*Maechler et al., 2022*)

applying the clusGap function with the *k-means* algorithm and 1,000 bootstraps. We also conducted a PERMANOVA (Permutational analysis of variance) using the vegan package (*Oksanen, 2015*) and the unweighted UniFrac distance to correlate differences in composition with altitude, latitude, or the clusters identified.

The overall microbiota composition was visualized with respect to relative abundance at the family taxonomic level. To determine the relationship between the abundance of the genera and latitude and altitude, we employed a linear model implemented in the Maaslin2 library (*Mallick et al., 2021*) with default arguments (method = LM, transform = LOG, normalization = TSS). To identify the enriched genera in the optimal clusters, we utilized the LEfSe program (*Segata et al., 2011*) using a cutoff of LDA > 2 and *p*-value < 0.05. All plots were prettified using the ggplot2 library (*Wilkinson, 2011*).

Additionally, we calculated the core microbiota of both the fly and the pulp to identify the most prevalent ASVs in the samples to gain insight into the most prevalent bacterial species. For this, we performed 1,000 random sub-samplings (rarefactions) with replacement and identified the ASVs present in 95% of the sub-samples and at least 75% of samples. Then, we performed a functional prediction with PICRUSt2 (*Douglas et al., 2019*) using the ASVs of the identified core microbiota. In the functional prediction, we searched for the presence and abundance of enzymes related to nitrogen fixation and the aminoacid biosynthetic pathways on the MetaCyc pathway classification and the EC (enzyme commission) number. We are fully aware of the fact that functional predictions obtained with PICRUSt2 are preliminary and need to be experimentally confirmed. But we believe that the patterns obtained can shed useful initial light into the phenomenon observed. Thus, we feel that the approach we followed is justified. It will also help design future confirmatory experiments, which were not conducted at this stage due to the large scope of the sampling process and the massive amount of data obtained in this first study, which is more ecological in nature.

## RESULTS

### Gut microbiota composition of *A. ludens*

From a total of 238 samples, 148 for the altitudinal and 90 for the latitudinal transect, we obtained 7,320,645 high-quality reads and 3,317 ASVs. Overall, taxonomic composition was different between the larvae and adults. Eight bacterial families dominated the larval microbiota with a mean abundance of over 1%, including Acetobacteraceae (37.9%), Rhizobiaceae (25.2%), Erwiniaceae (12.8%), Enterobacteriaceae (9.6%), Alcaligenaceae (5.4%), Lactobacillaceae (2.3%), Rhodanobacteraceae (1.6%), and Leuconostocaceae (1.5%). The gut microbiota in adults was similar between females and males, and dominated by the same three families, Enterobacteriaceae (44 and 41%), Rhizobiaceae (23 and 24%), and Pseudomonadaceae (8% and 7%). However, in females, these classes were followed in abundance by Alcaligenaceae (5.1%), Moraxellaceae (3.6%), Xanthomonadaceae (3.2%), Acetobacteraceae (3%), Morganellaceae (2.1%), Erwiniaceae (1.8%), and Lactobacillaceae (1%). On the other hand, in males we identified the families Alcaligenaceae (4.7%), Erwiniaceae (3.4%), Halomonadaceae (3.3%), Xanthomonadaceae (2.8%), Moraxellaceae (2%), and Lactobacillaceae (1%) (Figs. 2 and 3).

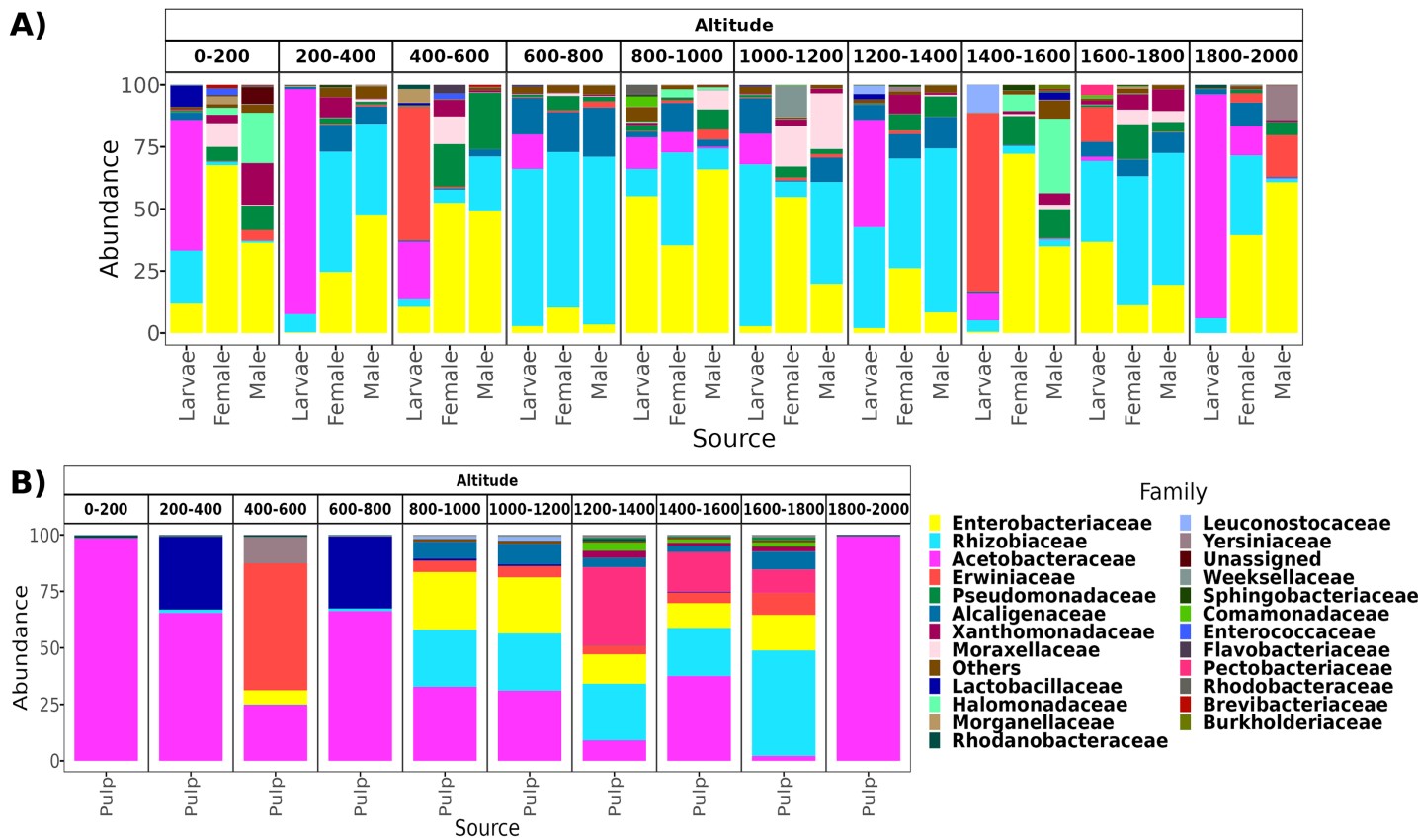

**Figure 2** **Relative abundance at the family taxonomic level of the gut microbiota of *A. ludens* in the altitudinal transect.** Gut microbiota of *A. ludens* (A) and pulp microbiota of *C.* x *aurantium* (B). The families with less than 5% of relative abundance were agglomerated in the "Others" category.

## Pulp microbiota composition of *C.* x *aurantium*

From a total of 73 final samples, we obtained 959,797 high-quality reads and 284 ASVs. The taxonomic composition was dominated by seven families with a mean abundance of over 1%, including Acetobacteraceae (56%), Rhizobiaceae (10%), Erwiniaceae (8.4%), Enterobacteriaceae (7.4%), Lactobacillaceae (7%), Pectobacteriaceae (4%), and Alcaligenaceae (2%) (see Figs. 2 and 3).

## Effect of latitude and altitude in the gut microbiota composition of *A. ludens*

The PCoA analysis based on the unweighted Unifrac distance metrics captured 22.2% and 27.8% of the variance within altitude and latitude, respectively (Fig. 4). The PERMANOVA analysis showed a correlation of the beta diversity of 2% ($R^2 = 0.02$, $F = 3.8$, $p$-value = 0.001) related to the altitude gradient and 3% ($R^2 = 0.03$, $F = 3.3$, $p$-value = 0.013) with the latitude gradient.

The genus-level abundance association analysis, using MaAsLin2, discerned 36 genera exhibiting statistically significant correlations: one with altitude and the remaining 35 with latitude (Fig. 5). The genus that correlated negatively with altitude was *Komagataeibacter*

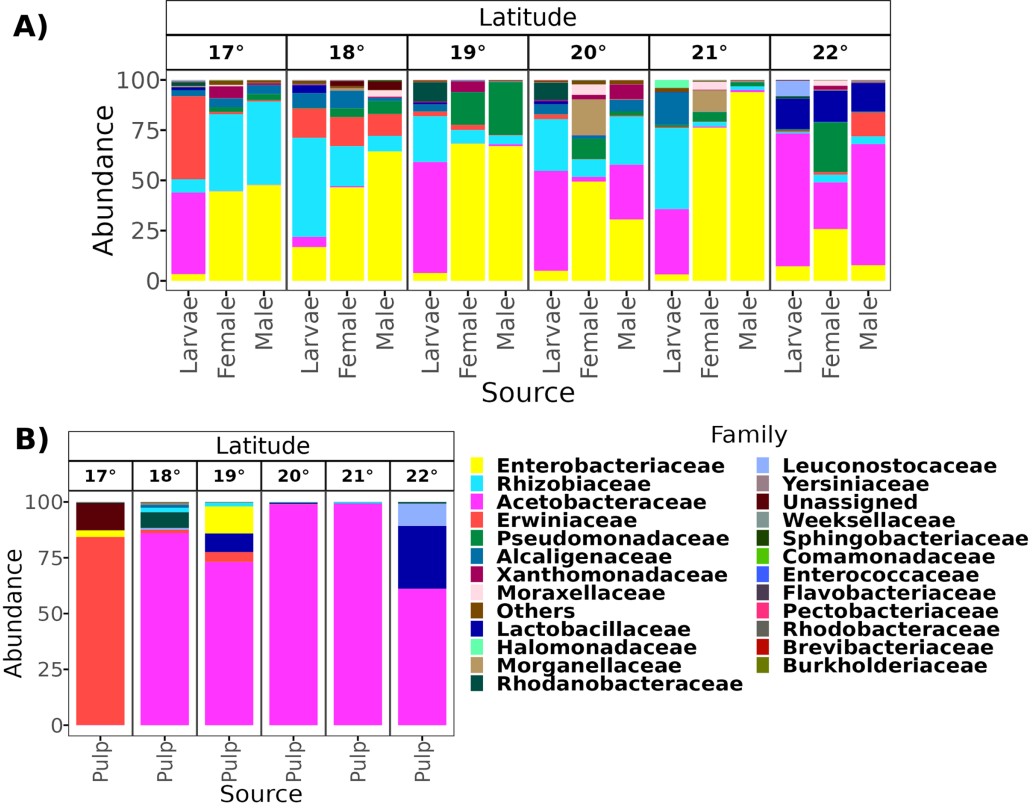

**Figure 3 Relative abundance at the family taxonomic level of the gut microbiota of *A. ludens* in the latitudinal transect.** The gut microbiota of *A. ludens* (A) and pulp microbiota of *C.* x *aurantium* (B). The families with less than 5% of relative abundance were agglomerated in the "Others" category.

in the larvae, but we underline the fact that the abundance was negligible (mean = 0.22%). Of the genera correlated with latitude, eleven had a mean over 1% of relative abundance: In larvae, one correlated positively (*Leuconostoc*) and three negatively (*Pantoea, Escherichia-Shigella, Tatumella*); in females, three correlated positively (*Gluconobacter, Acetobacter, Lactobacillus*) and four negatively (*Pantoea, Achromobacter, Phyllobacterium,* and *Kosakonia*); in males, three correlated positively (*Gluconobacter, Acetobacter,* and *Lactobacillus*) and one negatively (*Pseudomonas*).

## Determination of optimal altitudinal and latitudinal patterns

Determining clusters by *k-means* resulted in five and three clusters for the altitudinal and the latitudinal gradients, respectively (Fig. 4). The PERMANOVA analysis on the altitudinal clusters explained 24% of the variance ($R^2 = 0.24$, $F = 11.42$) and this seems driven by the fly developmental stage because samples in clusters were not differentiated by altitude but by fly developmental stages (Table S1). *Enterobacter* and *Phylobacterium* mainly drive the differences in the genera composition of these clusters (LDA > 5). The latitudinal clusters explained 20% of the variance ($R^2 = 0.20$, $F = 11.46$): the three clusters seem to show a transition between north and south locations because all developmental

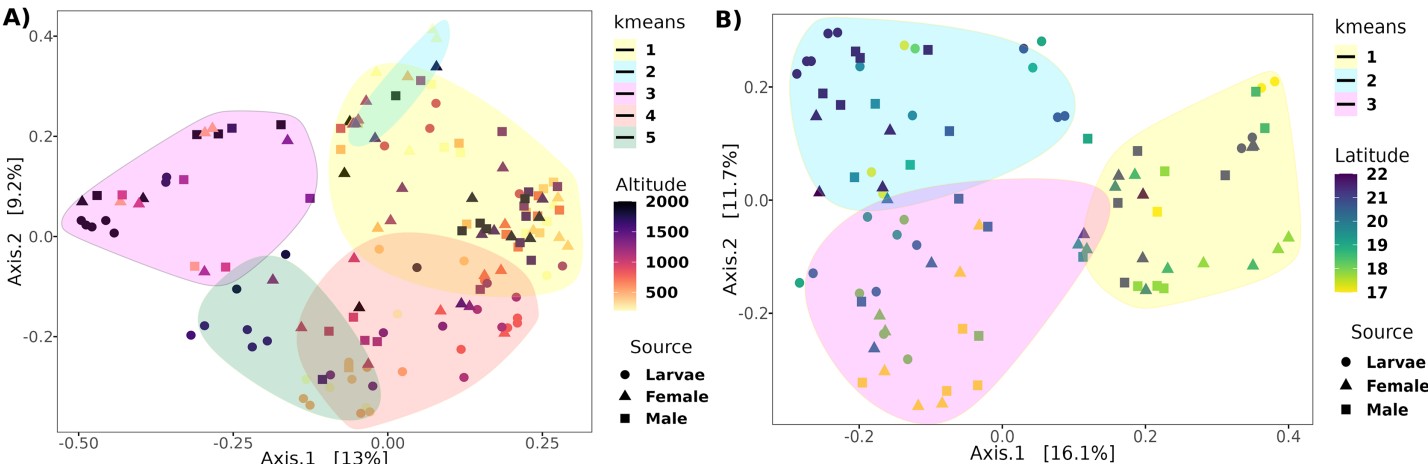

**Figure 4  Principal coordinates analysis of transects based on the unweighted UniFrac distance.** Ordination of samples for the (A) altitudinal and (B) latitudinal transect are shown. The *k-means* encircle the optimal clusters for each transect determined by the gap statistics.

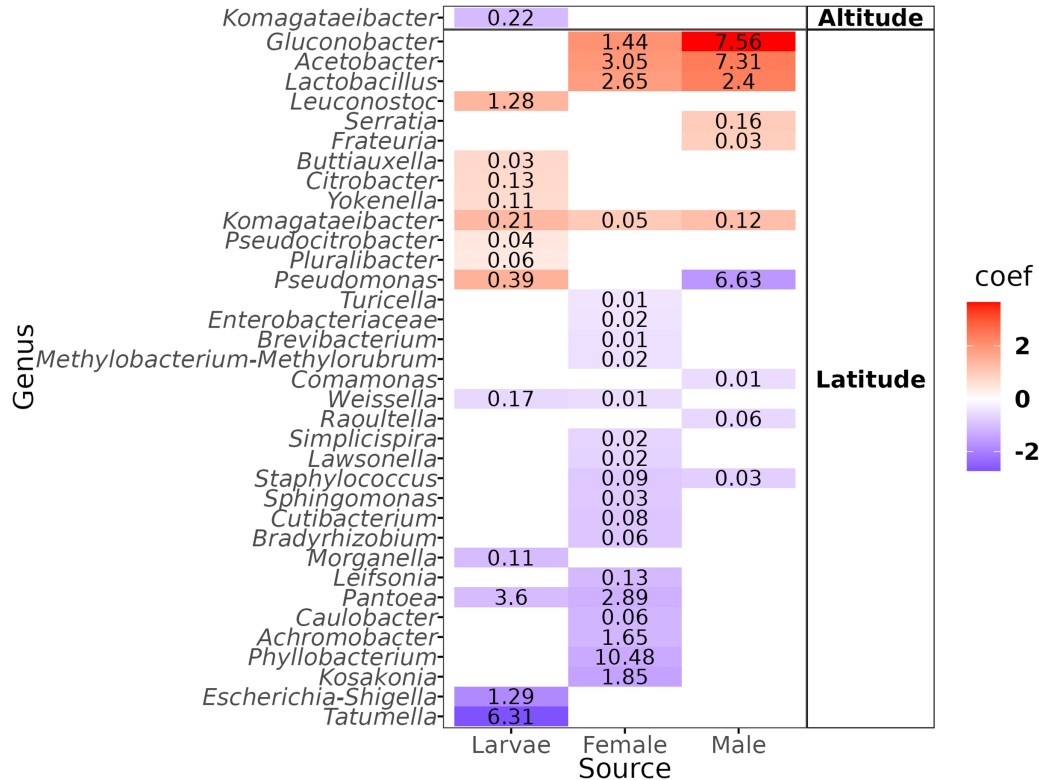

**Figure 5  Heatmap of the genera correlated with the altitude and latitude.** Color gradients represent the coefficients of the linear models obtained with Maaslin2. Numbers within the cell represent the mean relative abundance of the genera in the fly developmental stages.

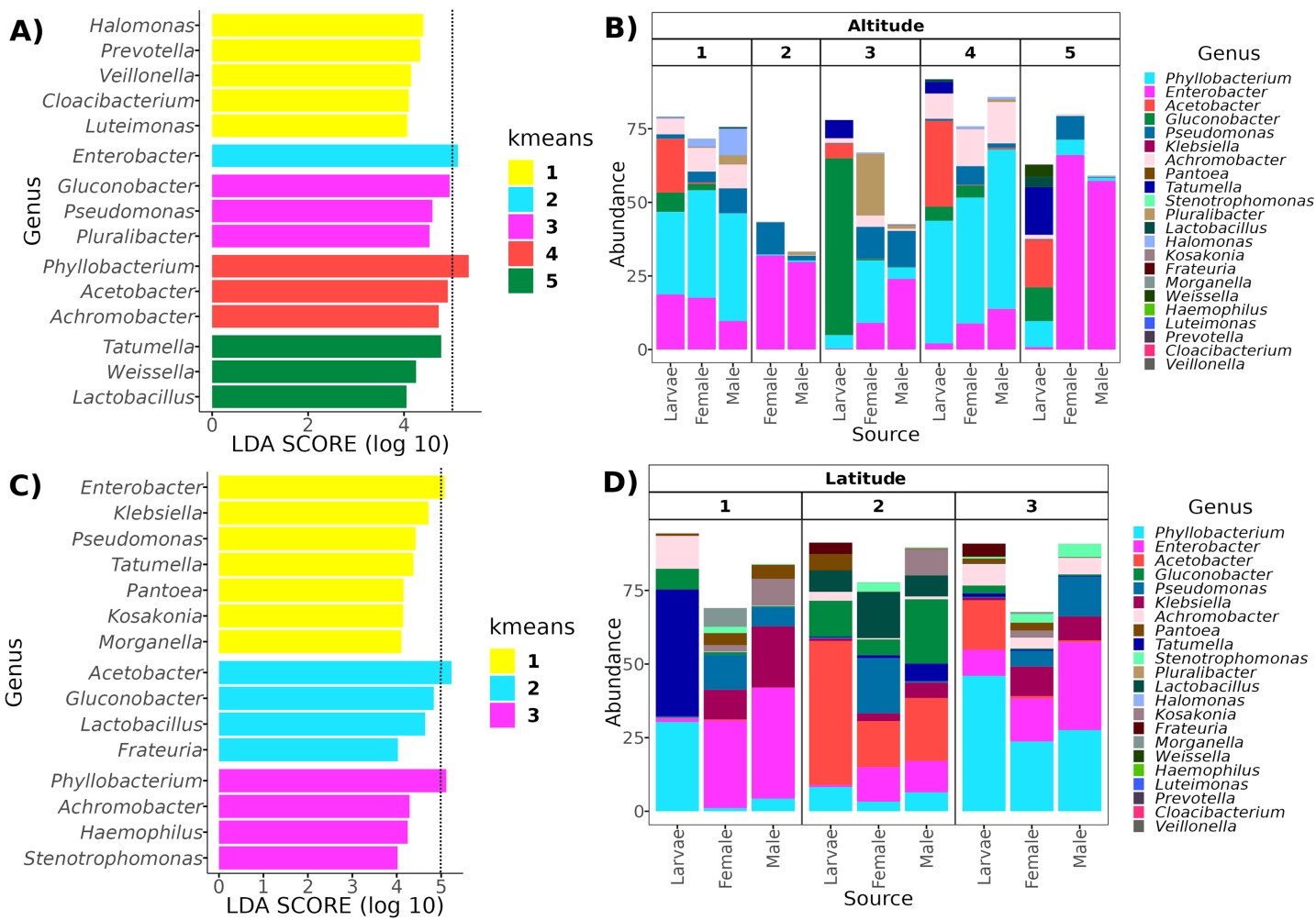

**Figure 6 Representative genera for each cluster identified with LEfSe.** In the top panels, genera enriched in the clusters along the altitudinal transect (A) and their relative abundance in each cluster and developmental stage (B). In the bottom panels, genera enriched in the clusters of the latitudinal transect (C) and their relative abundance in each cluster and developmental stage (D). The dotted line in (A) and (C) identifies the genera with an LDA > 5 (genera with the higher effect size).

stages are encompassed within them, but not with the altitude sample points (Table S2). *Acetobacter*, *Phyllobacterium*, and *Enterobacter* drive the differences in the genera composition of these clusters (LDA > 5) (see Fig. 6), when the first two are correlated with latitude, exhibiting a shift in composition, with an abundant presence of *Acetobacter* towards degree 22 and a greater presence of *Phyllobacterium* towards degree 17.

## The core microbiota of *A. ludens* and their functional role

The core microbiota of *A. ludens* comprised twelve ASVs corresponding to twelve species, but notably the retention of the core was different between larvae and adults (males and females) (Fig. 7). Seven species were exclusively found in the core microbiota of larvae: *Gluconobacter frateurii*, *G. cerevisiae*, *Phyllobacterium endophyticum*, *Acetobacter lambici*, *A. suratthaniensis*, *A. fabarum*, and *A. persici*. The exclusive species in the adult's core

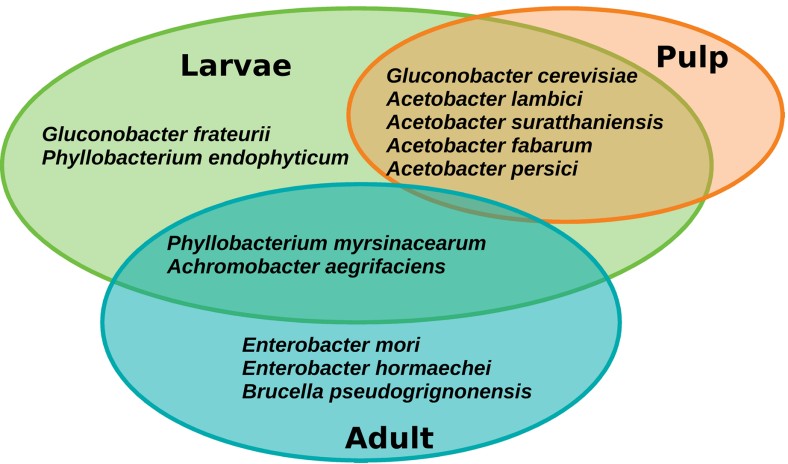

**Figure 7** **Venn diagram of the core microbiota of the gut of *A. ludens* (larvae and adults) and pulp of *C. x aurantium*.**

microbiota were *Enterobacter mori*, *E. hormaechei*, and *Brucella pseudogrignonensis*. Larvae and adults only share two core species: *P. myrsinacearum* and *Achromobacter aegrifaciens*. In contrast, the microbiota of the pulp of *C. x aurantium* was "leakier" than the microbiota of the fly, having no core at a prevalence threshold of 75%, only conserving five ASVs at 65% of prevalence. These five species were shared with the core microbiota of larvae, which were *G. cerevisiae*, *A. lambici*, *A. suratthaniensis*, *A. fabarum*, and *A. persici* (Fig. 7).

The PICRUSt2 pipeline calculated NSTI (nearest sequenced taxon index) values below 0.01 for nine core ASVs (Table S3). This means that these ASVs have a similarity of more than 98% to the reference genomes used by PICRUSt2 for functional prediction, representing the same species; thus, our prediction is reliable. These nine ASVs correspond to the species *G. cerevisiae*, *A. lambici*, *E. hormaechei*, *G. frateurii*, *B. pseudogridnonensis*, *E. mori*, *P. myrisinacearum*, *A. persici*, and *A. suratthaniensis*. The three-remaining core ASVs had NSTI values of 0.135, 0.033, and 0.055, which are below the PICRUSt2 cutoff of 2. Although they may not represent the same species in comparison with the reference genomes, they are very close.

The functional prediction resulted in a high proportion of potential biosynthetic pathways, with a relative abundance of 68% of pathways classified in the "Biosynthesis Ontological Category" (Fig. S1), followed by "Degradation/Utilization/Assimilation" pathways. As most core microbiota members of *A. ludens* are diazotrophs, we searched for predicted genes involved in nitrogen fixation and pathways for amino acid biosynthesis. The core microbiota of larvae was predicted to harbor ten proteins related to the assembly of nitrogenase (NifZ, NifX, NifT, NifQ, NifN, NifK, NifH, NifE, NifD, and NifB) and three protein regulators of nitrogen fixation (NifT, and fixK;CRP; FNR) (Fig. 8). In adults, proteins nifQ and nifU were present in all sample points, but some other proteins related to nitrogen fixation were not present at an altitude of 0–800 and 1,200–1,400 masl in males and 0–800 and 1,000–2,000 masl in females, and at latitude of 19° in females and 18° in

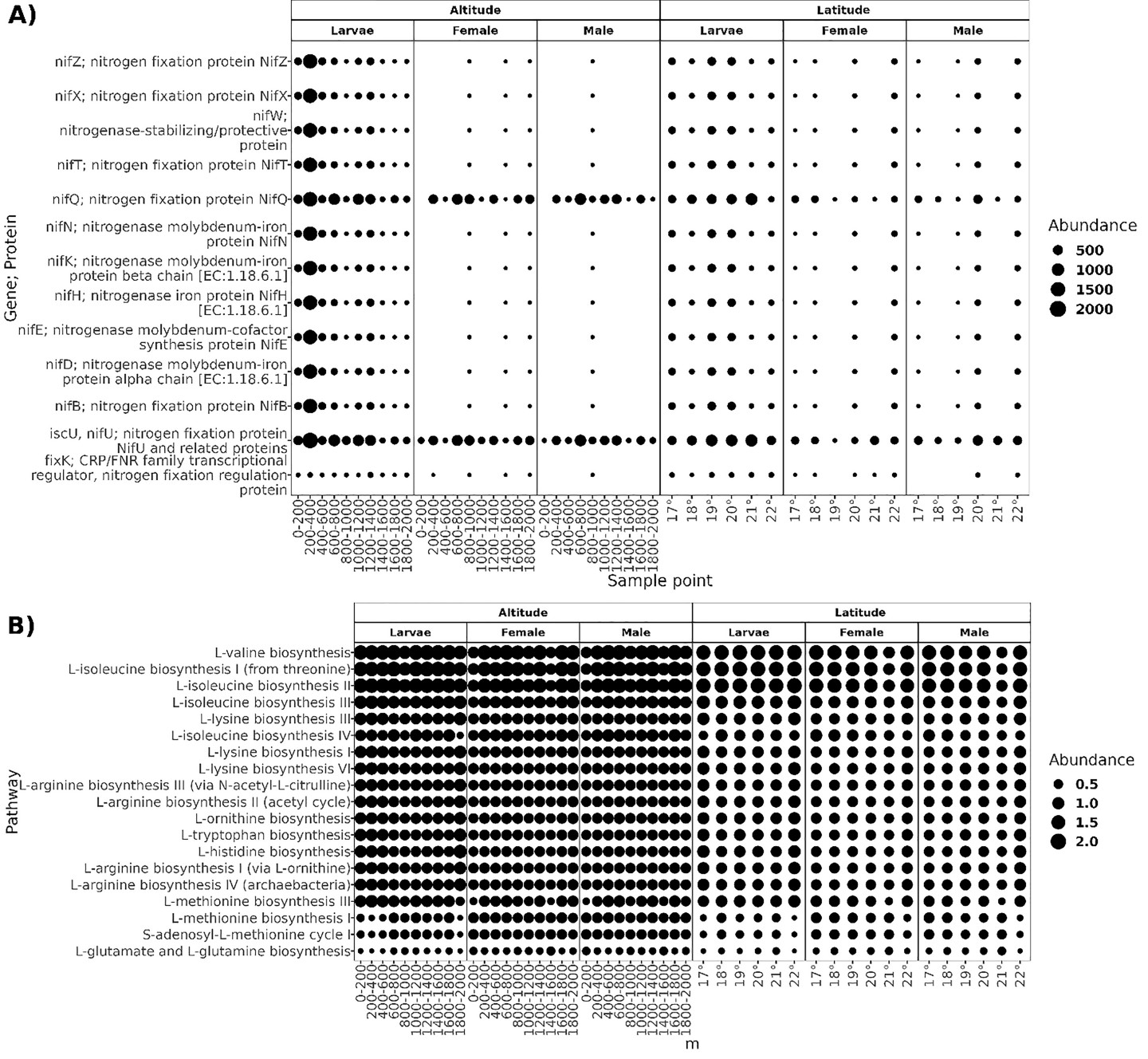

**Figure 8 Functional prediction of PICRUSt2.** (A) Genes; proteins related to the nitrogen fixation predicted in the core microbiota of larvae, females, and males; the size of the dots represents the abundance given in the predicted copy of gene per count of ASV. (B) Pathways of amino acid biosynthesis are present in all sample points in both transects; the dots' size represents the pathway's relative abundance.

males. The biosynthetic pathways of L-valine, L-isoleucine, L-lysine, L-arginine, L-ornithine, L-tryptophan, L-histidine, L-methionine, L-glutamate, and L-glutamine were present in all sampled conditions.

## DISCUSSION

In this study we analyzed the effect of altitude and latitude as well as the existence of a functional core on the gut microbiota of the notorious polyphagous pest *A. ludens* infesting a single fruit to control for host effect. We found that latitude exerted a stronger effect on microbiota diversity than altitude, with more associated variance on the microbiota composition. A trade-off between Acetobacteraceae and Rhizobiaceae drives the differences within the latitudinal clusters. Associated to this, we found that the core microbiota of *A. ludens* was comprised of twelve bacterial species, most of them potential NFB, with the most abundant belonging to the genera *Acetobacter* and *Phyllobacterium*. The core microbiota was preliminarily (*i.e.*, a confirmatory metagenomic study will possibly shed further light on this) predicted to be enriched with biosynthetic pathways with the potential capacity to perform BNF, and amino acid biosynthesis. These findings have important implications in our quest to understand the extreme polyphagy of this pestiferous fruit fly and to identify bacterial symbionts that naturally inhabit its gut to develop probiotics to improve mass rearing.

The core microbiota of *A. ludens* includes species within *Acetobacter* (*A. lambici*, *A. suratthaniensis*, *A. persici*, *A. fabarum*), *Gluconobacter* (*G. cerevisiae*, and *G. frateurii*), and *Phyllobacterium* (*P. myrsinacearum* and *P. endophyticum*). In previous studies, we found a relationship between the type of gut microbiota in the larvae of *A. ludens* and their wellbeing (*Aluja et al., 2021*; *Ochoa-Sánchez et al., 2022*). Larvae infesting occasional plant hosts rife with secondary chemicals (*e.g.*, *C. pubescens*) and non-natural hosts containing large amounts of polyphenols, particularly tannins (*e.g.*, *Psidium guajava* infested artificially), exhibited a severe dysbiosis characterized by a decrease in the abundance of bacteria within the Acetobacteraceae, yielding a deleterious phenotype with poor development. Part of this deleterious phenotype could be related to the depletion of bacteria belonging to the Acetobacteraceae family, which could supply nitrogen *via* the NFB to the larvae. We thus surmise that, the adequate development of larvae in this highly polyphagous pestiferous tephritid, depends on the presence of nitrogen-fixing bacteria. The composition of the core microbiota of *A. ludens* in our extensive survey here, covering many types of biomes, supports the hypothesis of a symbiotic relationship between *A. ludens* and nitrogen-fixing bacteria. The PICRUSt2 functional prediction preliminarily suggested that larvae seem to always harbor a microbiota that can perform BNF, regardless of the variation in the taxonomic composition, with a constant presence of the nitrogenase subunits nifK, nifH, and nifD, which have the catalytic activity (*Dean, Bolin & Zheng, 1993*; *Skøt, 2003*; *Zehr & Montoya, 2007*). It thus appears that *A. ludens* larvae can retain a diazotrophic functional group when infesting *C. x aurantium*, independent of the biome where trees grow, securing a bacterial-originated nitrogen supply.

The fruit pulp microbiota of *C.* x *aurantium* is dominantly composed of Acetobacteraceae, suggesting that the larvae acquired NFB from the pulp. However, it is important to note, as mentioned in the introduction, that females also vertically transmit bacteria *via* the eggs inserted into the pulp during oviposition (*Behar, Yuval & Jurkevitch, 2005*, *2008*). When larvae emerge, they start eating, moving, and defecating

(*Singh, Leppla & Adams, 1988*; *Carroll & Wharton, 1989*; *Ochoa-Sánchez et al., 2022*), producing a microorganism mediated decay in the fruit pulp, the "larval niche" *sensu Ochoa-Sánchez et al. (2022)*. The genera *Acetobacter*, *Gluconobacter* and *Phyllobacterium*, are common beneficial endophytes of plants supplying ammonia or synthesizing phytohormones such as indole acetic acid (IAA) (*Saravanan et al., 2008*; *Pedraza, 2016*; *Dwivedi, 2020*). However, *Acetobacter* and *Gluconobacter* also exhibit high respiratory activities and can metabolize sugars and pectins, producing a fermented environment by incomplete oxidation (*Flores-Encarnación et al., 1999*; *Macauley, McNeil & Harvey, 2001*; *Qi et al., 2014*; *Rani & Appaiah, 2012*). This fermentative activity could be triggered upon oviposition or the larva's hatch, producing the fruit's decay, as has been described in *A. ludens* (*Díaz-Fleischer & Aluja, 2003*) and the Medfly, *C. capitata* (*Behar, Jurkevitch & Yuval, 2008*). We surmise that at this point, beneficial bacteria of the plant become allied to larvae, producing a succession from fresh fruit to the "larval niche" (*i.e.*, decaying fruit), with the increase of available nitrogen within fruit by nitrogen fixation and amino acid biosynthesis from bacteria. This is supported by the functional prediction, where the core microbiota holds the pathway of L-glutamate and L-glutamine biosynthesis, containing the shuttle GS/GOGAT, which is the primary mechanism of ammonia assimilation (*Temple, Vance & Gantt, 1998*; *Muro-Pastor & Florencio, 2003*; *Hansen & Moran, 2011*). Besides, the pathway biosynthesis of L-valine, L-isoleucine, L-lysine, L-arginine, L-ornithine, L-tryptophan, L-histidine, and L-methionine also are present in the core microbiota, suggesting that the bacterial core have a biosynthetic role.

As mentioned before, *A. ludens* is a polyphagous species that can infest a wide range of fruits with high sugar and poor nitrogen content, representing a high C:N ratio diet (*Aluja et al., 1987*, *2000*, *2014*; *Birke, Acosta & Aluja, 2015*; *Birke & Aluja, 2018*). *Ochoa-Sánchez et al. (2022)* indicated that one of the two purported ancestral hosts, *C. edulis* (Rutaceae) is a "sugar bomb" based on its high carbohydrate/protein ratio (*Morton, 2013*; *Rivera, Bocanegra-García & Monge, 2010*; *Abo Taleb & Abdul Latif, 2023*). Along the same lines, the widely infested host bitter orange (*C.* x *aurantium*) represents another good example of the latter containing between 9.7–15.2 g of carbohydrates and 0.6–1.0 g of protein (*Cervoni, 2024*). Based on our results here, we confirm the findings and postulates of *Ochoa-Sánchez et al. (2022)* who wrote (under quotes as we cite *ad verbatim*): "we hypothesize that *A. ludens* lacks a phylogenetic-specific host-fly relationship but could rather exhibit a nutritional-specific host-fly relationship, capable of infesting any host with low toxicity and a high C:N ratio in nutrient composition".

Despite the high variation in the altitude transect sites, there was no strong correlation between beta diversity. Only 2% of the variance was correlated with the range of 0 to 2,000 masl, and only the extremely low abundant (mean abundance = 0.22%) genus *Komagataeibacter* was correlated negatively with altitude in larvae. This result is intriguing because the range of sampled altitude represents a wide range of mean temperatures, going from 26 °C (Jose Cardel) to 16 °C (Tlalnelhuayocan) (INEGI, *Sistema de Información Municipal*), encompassing tropical humid to humid temperate climates. It can be explained because the microenvironment within the fruit likely buffered the external environmental changes, conserving a fairly constant temperature and humidity (*Guillén*

*et al., 2022*). The analysis of the optimal clusters based on sample variance determined five clusters. However, the clusters reflect the differences in the developmental stages, not the altitude. The microbiota exhibited higher variances between larvae and adults than with altitude, which agrees with our previous report (*Aluja et al., 2021*). There is a bias towards adult samples in Clusters 1 and 2 but a bias towards larvae samples in Clusters 4 and 5. Thus, the retention of healthy microbiota at the entire sampled altitude transect is strong.

On the other hand, although the variance correlated with the latitude was similar to the one observed under altitude, the optimal clusters were not enriched by a specific developmental stage. Specifically, there is a transition based on latitude in the direction of Clusters 2, 1, to 3, where Cluster 2 contains more samples toward high latitudes (22°N) (Fig. S2), in contrast to Cluster 3, which contains more samples toward low latitudes (17°N). Interestingly, the LEfSe analysis shows that the genera with higher effects in the difference between these clusters were *Acetobacter* and *Phyllobacterium*, suggesting a transition between members of the Acetobacteraceae and Rhizobiaceae families in the guts of *A. ludens* third instar larvae and recently emerged adults (Fig. 6). This could be due to differences in the soil microbiota composition (and soil agricultural management) across the northern and southern latitudes where *C. x aurantium* is distributed, as the plant primarily acquires its microbiota from the soil-rhizosphere interface (*Trivedi et al., 2020*; *Banerjee & Van der Heijden, 2023*). But importantly, independent of the transition observed, the functional groups in the gut microbiome of *A. ludens* are maintained. Consequently, when we correlated the latitude with the most abundant genera, *Gluconobacter* and *Acetobacter* were correlated positively in females and males, and *Phyllobacterium* was correlated negatively in females. This observation suggests a trade-off between Acetobacteraceae/Rhizobiaceae, with Cluster 3 representing lower latitudes, having a higher significant abundance of members of Rhizobiaceae, in contrast to Cluster 2, with a higher abundance of Acetobacteraceae. It is worth noting that species within the genera *Acetobacter, Gluconobacter*, and *Phyllobacterium* have the ability for nitrogen fixation (*Boddey et al., 1991*; *Gonzalez-Bashan et al., 2000*; *Rasolomampianina et al., 2005*; *Kersters et al., 2006*; *Mantelin et al., 2006*; *Pedraza, 2008, 2016*; *Reis & Teixeira, 2015*; *Waller et al., 2019*).

Although adults harbor Acetobacteraceae, their core only retains the nitrogen fixer *P. myrsinacearum*. In a previous study, we found that during the metamorphosis from pupa to adult most of the Acetobacteraceae are eliminated (*Aluja et al., 2021*). This is consistent with *Mason, Auth & Geib (2023)*, who also found that bacterial populations declined upon adult emergence. We speculate that it is because of the obligatory requirement of oxygen of species within Acetobacteraceae, with adults only retaining bacteria able to survive in microaerophilic or anaerobic conditions (as *Enterobacter*). *Phyllobacterium* and many rhizobia species commonly inhabit plant nodules, where the regulation of oxygen availability is strict, reaching microaerobic conditions (*Wheatley et al., 2020*; *Rutten et al., 2021*), and therefore, could likely survive the metamorphosis process from larva to pupa to adult in the case of *A. ludens*, and remain in the gut of the adult stage. The larval requirements of nitrogen and other nutrients are more significant

than the ones by adults as they need to store energy for pupation (*Edgar, 2006*; *Nash & Chapman, 2014*; *Ongaratto et al., 2024*).

In conclusion, our results indicate that altitude has less influence than latitude on the microbiota of *A. ludens* when infesting the same host. The fruit may act as a refuge, maintaining stable temperature and humidity levels that are crucial for larval development. However, latitude significantly alters the microbiota composition, likely due to varying symbionts originating from the fruit, which in turn could be influenced by different soil conditions. Additionally, our results suggest that the potential for a geographic expansion of *A. ludens* is high. The extreme polyphagy of *A. ludens* requires a nutritional-specific host-fly interaction, that leads us to suggest that any fruit with a high C:N ratio whose pulp is inhabited by nitrogen fixing endophytes (or ones vertically transmitted by the females when inserting eggs into the pulp), and that contains low levels of polyphenols or other deleterious secondary metabolites, could potentially serve as an adequate host to this pest. This could have important practical implications worldwide as it could potentially help predict host range expansions caused by global warming or novel host relationships in areas where the pest is accidentally introduced.

## ACKNOWLEDGEMENTS

We fully recognize the extraordinary support of Gabriel A. Hernández Velásquez during the dangerous field collections, the technical support of Olinda Velázquez and Erick Enciso Ortiz in dissections of guts, of Yazmín Ríos Ibarra in the processing of DNA and amplicons, and the administrative support of Violeta Navarro Márquez, at INECOL.

### Funding

Daniel Cerqueda-García was supported by a CONAHCyT postdoctoral research fellowship (Estancias Posdoctorales por México 2022 (1)). This study was principally financed with resources from the Mexican Programa Nacional de Moscas de la Fruta (DGSV-SENASICA-SAGARPA (currently SADER)) *via* the Consejo Nacional Consultivo Fitosanitario (CONACOFI) through projects 41012-2018, 41013-2019, 80124-2020 and 80147-2021 awarded to Martín Aluja. Additional funds were provided by the Universidad de Valencia, Valencia, Spain *via* a Distinguished Professor Fellowship to Martín Aluja (UV-INV-EPC17-548793), and the Consejo Nacional de Ciencia y Tecnología (CONACyT)–Gobierno del Estado de Veracruz FOMIX grant (Project VER-2017-01-292397) to Martín Aluja. The project was also supported by the Generalitat Valenciana, Valencia, Spain (Project CIPROM/2021/042 to Andrés Moya). Finally, the logistical and financial assistance from the Instituto de Ecología, A.C. (INECOL) is recognized. The funders had no role in study design, data collection and analysis, decision to publish, or preparation of the manuscript.

## Grant Disclosures

The following grant information was disclosed by the authors:

CONAHCyT Postdoctoral Research Fellowship (Estancias Posdoctorales por México 2022 (1)).

Consejo Nacional Consultivo Fitosanitario (CONACOFI): 41012-2018, 41013-2019, 80124-2020 and 80147-2021.

Universidad de Valencia, Valencia, Spain via a Distinguished Professor Fellowship: UV-INV-EPC17-548793.

Consejo Nacional de Ciencia y Tecnología (CONACyT) – Gobierno del Estado de Veracruz FOMIX: Project VER-2017-01-292397.

Generalitat Valenciana, Valencia, Spain: Project CIPROM/2021/042.

Instituto de Ecología, A.C. (INECOL).

## Competing Interests

The authors declare that they have no competing interests.

## Author Contributions

- Martín Aluja conceived and designed the experiments, analyzed the data, authored or reviewed drafts of the article, and approved the final draft.
- Daniel Cerqueda-García analyzed the data, prepared figures and/or tables, authored or reviewed drafts of the article, and approved the final draft.
- Alma Altúzar-Molina performed the experiments, authored or reviewed drafts of the article, and approved the final draft.
- Larissa Guillén conceived and designed the experiments, authored or reviewed drafts of the article, and approved the final draft.
- Emilio Acosta-Velasco performed the experiments, prepared figures and/or tables, and approved the final draft.
- Juan Conde-Alarcón performed the experiments, authored or reviewed drafts of the article, and approved the final draft.
- Andrés Moya conceived and designed the experiments, authored or reviewed drafts of the article, and approved the final draft.

## Field Study Permissions

The following information was supplied relating to field study approvals (*i.e.*, approving body and any reference numbers):

As we worked with a notorious agricultural pest, neither ethical nor collection permits were necessary.

## DNA Deposition

The following information was supplied regarding the deposition of DNA sequences:

The raw data analyzed in this study is available at NCBI: PRJNA1007396.

https://www.ncbi.nlm.nih.gov/bioproject/PRJNA1007396/.

## Data Availability
The raw data analyzed in this study is available at NCBI: PRJNA1007396.

## Supplemental Information
Supplemental information for this article can be found online at http://dx.doi.org/10.7717/peerj.18555#supplemental-information.

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
