# Peer review of "Geographic variation and core microbiota composition of Anastrepha ludens (Diptera: Tephritidae) infesting a single host across latitudinal and altitudinal gradients"

_PeerJ, doi:10.7717/peerj.18555_

## Round 0.1 · original submission · Minor Revisions

A strong study, on the whole well written and of interest to a broad audience. There are a few places where citations are lacking and some sentences are too long and need to be broken into shorter sentences. Please follow all the reviewers' recommendations.

Reviewer 1 ·

Basic reporting

Geographical variation and core microbiota composition of Anastrepha ludens (Diptera: Tephritidae) infesting a single host across latitudinal and altitudinal gradients.
This study presents an intriguing investigation of the gut microbiota in Anastrepha ludens (adults and larvae) collected from Citrus x aurantium across various sites in Veracruz, spanning different altitudes and latitudes. The manuscript is well-written, and the study’s design and analysis are appropriately conducted. The results are both compelling and well-presented. I have some suggestions and would like to ask for a few clarifications regarding your manuscript.

Introduction
L91. A significant part of this paragraph discusses fruit domestication. However, the authors have not included references to studies that explore the effects of domestication on the reduction of secondary metabolites. I recommend improving this section by incorporating relevant studies in this area to provide a more comprehensive and consistent contextualization.
This study would benefit of a clear hypothesis regarding the expected findings on geographical variation in the composition or abundance of gut microbiota.

Materials and methods
When were fruit samples collected across all sampling sites? Were ripe fruit samples available simultaneously at all sites, regardless of geographical variation?

Discussion
L364 A comma (,) before ´with more associated variance on the microbiota composition´ could be useful.
There seems to be a lack of explanation for how latitude can influence gut microbiota.
Could Anastrepha ludens specimens from specific latitudes with higher gut microbiota content potentially impact other fruit hosts or exhibit more aggressive herbivory compared to those from other latitudes?
According the statement of the L39, I expected an explanation in the Discussion section regarding the potential implications of your findings for the mass-rearing of A. ludens.

Experimental design

It presents original research that falls within the journal's Aims and Scope, with a well-defined, relevant, and meaningful research question that addresses a clear knowledge gap. The methods are described in sufficient detail, providing enough information to allow for replication of the study.

Validity of the findings

Data are robust and statistically sound good. The conclusions are well-stated, directly linked to the original research question, and appropriately limited to the supporting results.

Reviewer 2 ·

Basic reporting

This article aims to investigate the possible effect of latitudinal and altitudinal gradients on microbiome of A. ludens, an important polyphagous pest in Mexico. Those fidings presents an important impact on basic and applied research on fruit flies and insects in general. The article is well writen and presents strong evidences to understand the insect-microbiota interactions.

However, a few points should be improved (detailed in the pdf attached).

1. The introduction is well-written but too extensive and should be more concise. Additionally, the main objective of this paper is to investigate the role of the latitudinal gradient on microbiota, a topic that the introduction does not currently address. It is important to include this make it more clear and to support the predictions made.

2. Material and Methods: No comment. Only few minor corrections.

3. Results. Only a few minor corrections.

4. Discussion: The authors discuss mostly aspects of the results findings, despite some too long sentences that make its hard to understand and some informations disconnected and not used for the argumentation. The authors also says in the end of abstract and discussion that those results could improove mas reared tecquiniques but nothing explaining on how those fidings could do that is addressed. The conclusion also should be rewrited since no answer on its original question is made.

Experimental design

No comment.

Validity of the findings

No comment.

Additional comments

No comment.

Annotated reviews are not available for download in order to protect the identity of reviewers who chose to remain anonymous.

Reviewer 3 ·

Basic reporting

The article is written in clear language, using appropriate technical terms.
The references cover the content addressed in the study and are sufficient for the development and arguments presented throughout the text.
The structure of the article is adequate, presenting figures and tables clearly, illustrating and presenting the results obtained.

Experimental design

The work is innovative and fits the scope of the journal, presenting clear objectives, with a well-conducted study that is innovative and offers important contributions to the field of insect ecology and pest management.
The methods are robust, featuring sophisticated bioinformatic analyses, and the results are clearly presented, contributing significantly to the understanding of interactions between insects and microbiota in different ecological contexts. The work is relevant to pest control and may open pathways for new integrated management approaches.

Validity of the findings

The scientific article is innovative and presents potential for the development of control strategies for the Mexican fruit fly, as it identifies the gut microbiota of both larvae and adults of this insect.
Environmentally friendly strategies aimed at pest control without environmental impact are extremely necessary, and studies like this elucidate many issues, providing opportunities for the development of new methods for insect pest control.
Therefore, I consider the work to be of great technical and academic relevance.
The authors present the data clearly and objectively, with solid results and a clear methodology, which can be replicated. They also provide interesting conclusions, highlighting the potential benefits of their results for future research.

Additional comments

Suggestions for revision:

Keywords: Remove Anastrepha ludens and Tephritidae. These words are already present in the title and do not need to be repeated in this section.

Line 129: Review the term "C. auratium."

Questions:
1. Which instar larvae were used?
2. Could the results have been different if the larvae were separated by instars?
3. Could larvae of different instars have different feeding/nutritional requirements?

As a suggestion for your future research, you could evaluate the nutritional requirements of different larval instars and the influence of gut microbiota at these stages.

I would like to congratulate you on the excellent research.

---

## Round 0.2 · Minor Revisions

While two reviewers have recommended we acept the manuscript, one reviewer has made a few edits to paragraph 3 of the introduction which improve its readability. Please implement those changes and resubmit a new version which will then be acceptable.

Reviewer 1 ·

Basic reporting

All my comments and suggestions were considered in the last version. I made some edits in the third paragraph of the introduction.

Experimental design

The experimental design was properly conducted

Validity of the findings

Ok

Additional comments

I would like to congratulate the authors for this excelent work.

Annotated reviews are not available for download in order to protect the identity of reviewers who chose to remain anonymous.

Reviewer 2 ·

Basic reporting

The authors improved the article as recomended by revisors. Then, it's ready for publication.

Experimental design

No comment.

Validity of the findings

No comment.

Additional comments

No comment.

Reviewer 3 ·

Basic reporting

Geographical variation and core microbiota composition of Anastrepha ludens (Diptera: Tephritidae) infesting a single host across latitudinal and altitudinal gradients.
This study presents an intriguing investigation of the gut microbiota in Anastrepha ludens (adults and larvae) collected from Citrus x aurantium across various sites in Veracruz, spanning different altitudes and latitudes. The manuscript is well-written, and the study’s design and analysis are appropriately conducted. The results are both compelling and well-presented. I have some suggestions and would like to ask for a few clarifications regarding your manuscript.

Experimental design

The work is innovative and fits the scope of the journal, presenting clear objectives, with a well-conducted study that is innovative and offers important contributions to the field of insect ecology and pest management.
The methods are robust, featuring sophisticated bioinformatic analyses, and the results are clearly presented, contributing significantly to the understanding of interactions between insects and microbiota in different ecological contexts. The work is relevant to pest control and may open pathways for new integrated management approaches.

Validity of the findings

The scientific article is innovative and presents potential for the development of control strategies for the Mexican fruit fly, as it identifies the gut microbiota of both larvae and adults of this insect.
Environmentally friendly strategies aimed at pest control without environmental impact are extremely necessary, and studies like this elucidate many issues, providing opportunities for the development of new methods for insect pest control.
Therefore, I consider the work to be of great technical and academic relevance.
The authors present the data clearly and objectively, with solid results and a clear methodology, which can be replicated. They also provide interesting conclusions, highlighting the potential benefits of their results for future research.

---

## Round 0.3 · accepted · Accept

All reviewers' comments have been addressed and the manuscript is now ready for publication.